# Young Age, Female Sex, and No Comorbidities Are Risk Factors for Adverse Reactions after the Third Dose of BNT162b2 COVID-19 Vaccine against SARS-CoV-2: A Prospective Cohort Study in Japan

**DOI:** 10.3390/vaccines10081357

**Published:** 2022-08-19

**Authors:** Ryuta Urakawa, Emiko Tanaka Isomura, Kazuhide Matsunaga, Kazumi Kubota

**Affiliations:** 1Department of Pharmacy, Osaka University Dental Hospital, 1-8 Yamada-oka, Suita 565-0871, Osaka, Japan; 2Department of Clinical Pharmacy Research and Education, Graduate School of Pharmaceutical Sciences, Osaka University, 1-6 Yamada-oka, Suita 565-0871, Osaka, Japan; 3First Department of Oral and Maxillofacial Surgery, Graduate School of Dentistry, Osaka University, 1-8 Yamada-oka, Suita 565-0871, Osaka, Japan; 4Second Department of Oral and Maxillofacial Surgery, Graduate School of Dentistry, Osaka University, 1-8 Yamada-oka, Suita 565-0871, Osaka, Japan; 5Department of Healthcare Information Management, The University of Tokyo Hospital, 7-3-1 Hongo, Bunkyo-ku, Tokyo 113-8655, Japan

**Keywords:** infectious disease, booster, vaccination interval, adverse event, side effect

## Abstract

Background: This study compared the adverse events (AEs) of the second and third doses of BNT162b2, as well as investigated the impact of vaccine recipients’ background and vaccination interval on the AEs of the third dose. Methods: We conducted a questionnaire survey of AEs among health care workers at Osaka University Dental Hospital. Chi-square tests were performed to compare AEs to the administration of second and third vaccine doses. Logistic regression analyses were conducted to identify factors influencing the presence of AEs using age, sex, comorbidities, and the vaccination interval. Spearman’s rank correlation coefficient was calculated to investigate the correlation between age, vaccination interval, and severity of each AE. Results: The third dose of BNT162b2 was associated with significantly more frequent or milder AEs than the second dose. Logistic regression analyses detected significant differences in six items of AEs by age, three by sex, two by comorbidities, and zero by vaccination interval. Consistently, the risk of AEs was greater among younger persons, females, and those without comorbidities. Significant negative correlations were detected between age and vaccination interval, and between age and the severity of most AEs. Conclusions: Young, female, and having no comorbidities are risk factors for AEs after the third dose of BNT162b2, while vaccination interval is not.

## 1. Introduction

Coronavirus disease 2019 (COVID-19) is an infectious disease caused by severe acute respiratory syndrome coronavirus 2 (SARS-CoV-2), which was declared a pandemic by the WHO in March 2020 [1] and has since spread worldwide and continues to cause many patient deaths. Because of its high infectivity and high rates of severe disease and mortal-ity, COVID-19 has had numerous medical, economic, and lifestyle impacts in many parts of the world. COVID-19 is highly contagious compared with seasonal influenza [2] and is creating waves of infection in many countries with repeated mutations. The emergence of variants such as alpha, beta, gamma, delta, and omicron strains, and especially the emergence of strains such as delta and omicron, which are more infectious and less inhibited by vaccines than conventional strains [3], is a cause for concern. In general, the more the virus replicates, the greater the risk that a variant will emerge, and the higher the risk that more dangerous or more easily transmitted variants may emerge in the future. The mechanism of viral mutation is determined by multiple virus- and host-dependent processes. RNA viruses mutate faster than DNA viruses, single-stranded viruses mutate faster than double-stranded viruses, and genome size is negatively correlated with mutation rate [4]. Since SARS-CoV-2 is a positive-strand RNA virus of relatively large size, we should continue to watch for rapid mutation [5].

SARS-CoV-2 is transmitted mainly through human saliva by droplet, aerosol, and contact transmission [6]. While hand hygiene, wearing masks, ventilating rooms, and keeping a safe distance from others are effective means of preventing infection [6], vaccination is a very useful means of acquiring immunity against SARS-CoV-2 and preventing the establishment of infection. Vaccination with two doses of vaccine can prevent the onset or severity of disease due to the induction of antibodies against SARS-CoV-2 and memory B and T cells [7]. However, 5 or 6 months after vaccination, the effectiveness in preventing infection decreases to 47% [8], possibly due to a decrease in antibody titer over time [9,10]. Therefore, a third booster vaccination is being promoted in many countries, and additional vaccinations have been reported to be effective in preventing hospitalization or severe disease caused by COVID-19 and reducing mortality [11,12,13]. Since dangerous variants of SARS-CoV-2 may appear and create waves of infection in the future, additional vaccinations may be needed. Administration of a fourth dose of vaccine began in Japan in May 2022.

The frequency of adverse reactions to the vaccine is very high for injection site reactions, fatigue, and headache, followed by myalgia, chills, joint pain, and fever. It has been reported that adverse reactions occur more frequently and are more severe after the second dose than after the first, in females than in males, and in younger than in older vaccine recipients [14,15], which we previously described with a multivariate analysis report [16]. These may be explained by previous reports of sex and age differences in the immune system. Sex hormones affect immune responses via secondary metabolites that bind to receptors such as estrogen receptors (ERs) and peroxisome proliferator-activated receptors (PPARs) [17]. Aging brings age-related changes in the cells of the immune system, the soluble molecules that guide the maintenance and function of the immune system, and the lymphoid organs that coordinate both the maintenance of lymphocytes and the initiation of immune responses [18]. There is still room for debate as to whether the effects of age and sex are similar for the third and subsequent booster vaccinations, which are expected to be given continuously in the future, and what the effects of vaccination intervals are. Reports on the efficacy [11,12,13] and safety [12,13] of the third dose of vaccine continue to be published.

Based on the hypothesis that the incidence and severity of adverse reactions to vaccines vary with the second and third doses and their intervals, as well as with the background of the vaccine recipients, this study used multivariate analysis to identify factors that contribute to adverse reactions to vaccines.

## 2. Materials and Methods

### 2.1. Research Population

Health care workers at Osaka University Dental Hospital who received the third dose of the BNT162b2 vaccine were included in the research. Adverse reactions for the second dose were based on data collected in our previous study [16], and adverse reactions for the third dose were collected through a questionnaire between January 2022 and March 2022. Those who did not indicate their consent in the research consent section of the questionnaire or those with incomplete forms were excluded from the study. Additionally, those vaccinated with SARS-CoV-2 vaccines other than BNT162b2 were excluded. A total of 617 health care workers were vaccinated with the third dose, of whom 224 consented to the study and completed the questionnaire (response rate: 36.3%). There were 194 valid responses (valid response rate: 86.6%), of which 175 who were vaccinated with all three doses of BNT162b2 were included in this study for the third vaccination.

### 2.2. Questionnaire Survey for the Third Vaccination

The questionnaire was anonymous and consisted of questions regarding study consent, age, sex, vaccination date, type of second and third vaccination, comorbidities, and type and severity of adverse reactions. Comorbidities consisted of cardiac disease, renal disease, hepatic disease, hematologic diseases, blood coagulation abnormalities, immunodeficiency, and others. The types of adverse reactions consisted of injection site reaction, fatigue, chills, fever, arthralgia, myalgia, headache, diarrhea, nausea, and vomiting. The severity of adverse reactions was evaluated based on the Common Terminology Criteria for Adverse Events (CTCAE) Version 5.0. The severity of adverse reactions that did not develop was defined as Grade 0. The most severe grade of all classified adverse reactions was defined as the worst adverse reaction grade.

### 2.3. Statistical Analysis

Chi-square tests were performed to compare the adverse reactions of the second and third doses in two ways: a two-group comparison of Grade 0 and Grade 1–3 to examine differences in the presence or absence of adverse reactions, and a two-group comparison of Grade 1 and Grade 2, 3 to examine differences between mild and moderate to severe reactions. Univariate and multivariate logistic regression analyses were performed to identify factors affecting the presence of each adverse reaction, with age, sex, presence of comorbidities, and the interval between the second and third vaccinations as independent variables and the presence of each adverse reaction as a dependent variable. Spearman’s rank correlation coefficient was calculated to investigate the correlation between age, vaccination interval, and severity of each adverse reaction. All statistical analyses were performed using IBM SPSS Statistics 26, and *p* < 0.05 on a two-tailed test was defined as a significant difference.

### 2.4. Ethical Review

This research was approved by the Ethical Review Committee of Osaka University Dental Hospital with the approval number of R3-E11-2.

## 3. Results

### 3.1. Profiles of Research Subjects and Adverse Reactions of Second and Third Doses

Table 1 shows the profiles of the subjects and Table 2 shows the severity of each adverse reaction at the time of the second and third doses. No severe AEs such as myocarditis, cardiovascular diseases, and thrombosis developed.

### 3.2. Comparison of Adverse Reactions between Second and Third Dose

The results of the chi-square test are shown in Table 3 and Table 4. Table 3 shows that occurrence of injection site reaction, fatigue, chills, arthralgia, myalgia, nausea, and any AE were significantly more frequent with the third dose. Table 4 shows that there were significantly more mild cases in the second dose for fever and in the third dose for fatigue, arthralgia, myalgia, headache, and any AE.

### 3.3. Factors Affecting the Presence of Each Adverse Reaction of Third Dose

The results of the logistic regression analysis are shown in Table 5. In the univariate model, significant differences were detected in six adverse reactions (injection site reaction, chills, fever, arthralgia, headache, and any AE) by age, six adverse reactions (injection site reaction, fatigue, arthralgia, myalgia, headache, and any AE) by sex, and three adverse reactions (chills, fever, and any AE) by comorbidities. In the multivariate model, significant differences were observed in six adverse reactions (injection site reaction, chills, fever, headache, diarrhea, and any AE) by age, three (fatigue, myalgia, and headache) by sex, and two (chills and fever) by comorbidities. Consistently, the risk of developing adverse reactions was greater among the young, females, and those vaccinated without comorbidities. As for the vaccination intervals, no significant differences were detected in both univariate and multivariate models.

### 3.4. Correlation between Age, Vaccination Interval, and Severity of Each Adverse Reaction

Correlation coefficients between age and vaccination interval and severity of adverse reactions are shown in Table 6. Younger age was significantly correlated with the severity of many adverse reactions (injection site reaction, fatigue, chills, fever, arthralgia, headache, and the worst adverse reaction). Only myalgia showed a weak correlation between vaccination interval and severity of adverse reactions, but no other significant correlation was observed. A significant negative correlation was detected between age and vaccination interval.

## 4. Discussion

Vaccinations against SARS-CoV-2 in Japan are currently approved and underway for BNT162b2, mRNA-1273, AZD1222, and NVX-CoV2373. The third dose began in December 2021 for those 12 years of age and older. As of May 1, approximately 80% of the population had received the second dose and 50% had completed the third dose [19]. In this study, the study subjects were vaccine recipients of BNT162b2, which is the earliest to obtain regulatory approval and one of the most used vaccines against SARS-CoV-2 in Japan.

The response rate in the questionnaire after the third dose was much lower than that of the previous study [16] (55.6%). The reasons for this may be attributed to the fact that the second dose was given on a day within a 5-day vaccination period, while the third dose could be given on any day 6 months or more after the second dose, and that the vaccination location was changed from one designated location to an any location. Although it was possible to increase the sample size by devising a questionnaire method, we believe that a sufficient number were collected for analysis. The results of the chi-square test for the second and third doses showed that AEs after the third dose of BNT162b2 vaccine were significantly more frequent or significantly milder than AEs after the second dose of BNT162b2 vaccine. The CDC reports that in various symptoms seen within 1 week after the booster vaccination, local and systemic reactions and health impacts were less frequent than with the second dose of the primary series [20]. While other reports have indicated that the incidence of adverse reactions for the second and third doses was similar [21], the results of this study are different. This may be due to racial differences, as this study was conducted on Japanese health care workers [22].

Regarding age, logistic regression analysis showed that many adverse reactions were more likely to occur at younger ages. Spearman’s correlation coefficient also showed that the younger the recipient of the vaccine, the greater the severity of the disease. Regarding sex, the results of the logistic regression analysis showed that females were more likely to have adverse reactions than males. These results are consistent with our previous study of adverse reactions after the first and second vaccination [16] and with other reports of immune responses [23]. Regarding comorbidities, fever and chills occurred significantly more frequently in vaccinated subjects without comorbidities. We think this reflects a report that vaccinated individuals with comorbidities have reduced antibody production after mRNA-based Pfizer-BioNTech vaccination [23]. Notarte et al. studied the humoral immune response after vaccination and noted that older adults produce lower levels of anti-SARS-CoV-2 antibody production, and females appear to be associated with higher antibody production due to estrogen and the immunomodulatory properties of the X chromosome [23]. The results of this study clinically reflect the report of Notarte et al. and indicate that side effects may be stronger with increased amounts of antibody.

Although there was a significant weak correlation between vaccination interval and severity of adverse reactions only for myalgia, when combined with the results of the logistic regression analysis, we conclude that this may be a significant difference by chance. Remarkably, a significant negative correlation was detected between age and vaccination interval, with younger people having longer vaccination intervals. At Osaka University Dental Hospital, the first and second doses were given as group vaccinations, so the maximum difference was only 5 days, while the third dose could be given at any date and time. Therefore, this result may reflect either a willingness of the elderly to be vaccinated, or a hesitation of younger people to receive the third dose due to the severity of the adverse reactions of the second dose. Based on the results of this study, an interval of at least 6 months between vaccinations is not associated with the risk or severity of adverse reactions. In addition, it has been reported that the immune response after the second vaccination decreases over time [24], so it is recommended that the vaccine be administered before its effectiveness wears off.

There are two limitations of this study. The first is selection bias. Since the study is based on reports by health care workers at a single hospital in Japan, differences with non-medical workers and racial differences are possible. In addition, since the second and third dose cases are not paired, there may be a selection bias in that vaccinated recipients with severe symptoms at the second dose are not vaccinated at the third dose. Therefore, one of the results of this study, that the third dose was associated with a higher frequency and lower severity of adverse reactions than the second dose, may be coincidental. It is desirable that similar studies be conducted in the future in many regions, including Japan, using paired second- and third-dose recipients. The second limitation is that we did not measure the level of immune response, such as antibody titers. In this study, younger participants were found to have longer vaccination intervals than older participants. Given that younger people have higher immune responses, it is possible that there is an appropriate age-specific vaccination interval.

Three years have passed since COVID-19 became a worldwide concern, and the pandemic shows no signs of ending. Information on the efficacy of vaccines against SARS-CoV-2 and its adverse reactions is abound in various places. While continued booster vaccination is likely to be necessary in the future, strategic promotion of vaccination based on accurate information will continue to be necessary, and this study provides one of the key findings. In the future, it may be possible to propose a vaccination interval for the SARS-CoV-2 vaccine according to age, sex, and comorbidities in terms of antibody titer, effectiveness in preventing infection, and severity of adverse reactions. We also expect that accumulation of evidence such as that of our study on the SARS-CoV-2 vaccine will promote vaccination based on correct knowledge and understanding of the vaccine throughout the world.

## 5. Conclusions

In response to our hypothesis that the incidence and severity of adverse reactions to vaccines vary with the second and third doses and their intervals, as well as with the background of the vaccine recipients, we found that the frequency and severity of adverse reactions were higher in females and young adults, and may be higher in recipients with no comorbidities. On the other hand, we found that immunization interval was not related to adverse reactions. It is recommended that BNT162b2 vaccination be strategically promoted according to the background and vaccination history of the recipient.

## Figures and Tables

**Table 1 vaccines-10-01357-t001:** Demographic characteristics of the research population.

	Second Dose (N = 458)	Third Dose (N = 175)
Mean age (SD, range)		
All	38.9 (12.8, 19–77)	45.9 (13.2, 20–75)
Sex		
Male number (%)	195 (42.6)	71 (40.6)
Female number (%)	263 (57.4)	104 (59.4)
Comorbidities		
Yes (%)	54 (11.8)	38 (21.7)
No (%)	404 (88.2)	137 (78.3)
Days between vaccinations (SD, range)	-	242.3 (19.3, 183–322)

Comorbidities included the presence of cardiac disease, renal disease, hepatic disease, hematologic diseases, blood coagulation abnormalities, and immunodeficiency, etc. Days between vaccinations showed vaccination interval between the second dose and the third dose of BNT162b2.

**Table 2 vaccines-10-01357-t002:** Severity of each adverse reaction after the second and third doses of BNT162b2 vaccination.

Severity	Grade 0	Grade 1	Grade 2	Grade 3
	Second Dose/Third Dose	Second Dose/Third Dose	Second Dose/Third Dose	Second Dose/Third Dose
	n (%)	n (%)	n (%)	n (%)
Injection site reaction	168 (36.7)/22 (12.6)	190 (41.5)/90 (51.4)	95 (20.7)/55 (31.4)	5 (1.1)/8 (4.6)
Fatigue	201 (43.9)/47 (26.9)	90 (19.7)/65 (37.1)	139 (30.3)/47 (26.9)	28 (6.1)/16 (9.1)
Chills	389 (84.9)/125 (71.4)	48 (10.5)/38 (21.7)	21 (4.6)/12 (6.9)	0/0
Fever	346 (75.5)/130 (74.3)	102 (22.3)/35 (20.0)	10 (2.2)/10 (5.7)	0/0
Arthralgia	318 (69.4)/105 (60.0)	67 (14.6)/45 (25.7)	61 (13.3)/18 (10.3)	12 (2.6)/7 (4.0)
Myalgia	361 (78.8)/96 (54.9)	48 (10.5)/56 (32.0)	41 (9.0)/17 (9.7)	8 (1.7)/6 (3.4)
Headache	266 (58.1)/87 (49.7)	79 (17.2)/51 (29.1)	90 (19.7)/30 (17.1)	23(5.0)/7 (4.0)
Diarrhea	439 (95.9)/168 (96.0)	16 (3.5)/7 (4.0)	2 (0.4)/0	1 (0.2)/0
Nausea	407 (88.9)/143 (81.7)	23 (5.0)/20 (11.4)	27 (5.9)/7 (4.0)	1 (0.2)/5 (2.9)
Vomiting	447 (97.6)/173 (98.9)	9 (2.0)/1 (0.6)	1 (0.2)/1 (0.6)	1 (0.2)/0
Worst adverse reaction	54 (11.8)/9 (5.1)	152 (33.2)/79 (45.1)	204 (44.5)/67 (38.3)	48 (10.5)/20 (11.4)

Grade refers to the severity of each adverse reaction with reference to CTCAE version 5.0. The most severe grade of all classified adverse reactions was defined as the worst adverse reaction grade.

**Table 3 vaccines-10-01357-t003:** Occurrence of adverse reactions in the second and third doses of BNT162b2 vaccination.

Severity	Grade 0	Grade 1–3	χ^2^ (df = 1)	*p* Value
	Second Dose/Third Dose	Second Dose/Third Dose		
	n (%)	n (%)		
Injection site reaction	168 (36.7)/22 (12.6)	290 (63.3)/153 (87.4)	35.04	<0.01
Fatigue	201 (43.9)/47 (26.9)	257 (56.1)/128 (73.1)	15.41	<0.01
Chills	389 (84.9)/125 (71.4)	69 (15.1)/50 (28.6)	15.13	<0.01
Fever	346 (75.5)/130 (74.3)	112 (24.5)/45 (25.7)	0.11	0.74
Arthralgia	318 (69.4)/105 (60.0)	140 (30.6)/70 (40.0)	5.08	0.02
Myalgia	361 (78.8)/96 (54.9)	97 (21.2)/79 (45.1)	36.22	<0.01
Headache	266 (58.1)/87 (49.7)	192 (41.9)/88 (50.3)	3.59	0.06
Diarrhea	439 (95.9)/168 (96.0)	19 (4.1)/7 (4.0)	0.01	0.93
Nausea	407 (88.9)/143 (81.7)	51 (11.1)/32 (18.3)	5.68	0.02
Vomiting ^a^	447 (97.6)/173 (98.9)	11 (2.4)/2 (1.1)	-	0.53
Worst adverse reaction	54 (11.8)/9 (5.1)	404 (88.2)/166 (94.9)	6.24	0.01

Chi-square tests were performed to compare occurrence of adverse reactions in the second and third doses. ^a^ Chi-square value was omitted and *p* value showed result of the Fisher’s exact test. The most severe grade of all classified adverse reactions was defined as the worst adverse reaction grade.

**Table 4 vaccines-10-01357-t004:** Comparison of second and third doses of BNT162b2 for mild and moderate to severe adverse reactions.

Severity	Grade 1	Grade 2–3	χ^2^ (df = 1)	*p* Value
	Second Dose/Third Dose	Second Dose/Third Dose		
	n (%)	n (%)		
Injection site reaction	190 (65.5)/90 (58.8)	100 (34.5)/63 (41.2)	1.93	0.17
Fatigue	90 (35.0)/65 (50.8)	167 (65.0)/63 (49.2)	8.83	<0.01
Chills	48 (69.6)/38 (76.0)	21 (30.4)/12 (24.0)	0.60	0.44
Fever	102 (91.1)/35 (77.8)	10 (8.9)/10 (22.2)	5.10	0.02
Arthralgia	67 (47.9)/45 (64.3)	73 (52.1)/25 (35.7)	5.06	0.02
Myalgia	48 (49.5)/56 (70.9)	49 (50.5)/23 (29.1)	8.25	<0.01
Headache	79 (41.1)/51 (58.0)	113 (58.9)/37 (42.0)	6.86	<0.01
Diarrhea ^a^	16 (84.2)/7 (100.0)	3 (15.8)/0 (0.0)	-	0.54
Nausea	23 (45.1)/20 (62.5)	28 (54.9)/12 (37.5)	2.39	0.12
Vomiting ^a^	9 (81.8)/1 (50.0)	2 (18.2)/1 (50.0)	-	0.42
Worst adverse reaction	152 (37.6)/79 (47.6)	252 (62.4)/87 (52.4)	4.85	0.03

Chi-square tests were performed to examine differences between mild (Grade 1) and moderate to severe (Grade 2–3) adverse reactions. ^a^ Chi-square value was omitted and *p* value showed result of fisher’s exact test. The most severe grade of all classified adverse reactions was defined as the worst adverse reaction grade.

**Table 5 vaccines-10-01357-t005:** Influence of age, sex, comorbidities, and vaccination interval on the presence of adverse reactions after the third dose of BNT162b2 vaccination.

	Age (10+)	Sex (Male (Ref.)/Female)	Comorbidities (with (Ref.)/without)	Vaccination Interval (1+)
	OR (95%CI, Wald χ^2^)	OR (95%CI, Wald χ^2^)	OR (95%CI, Wald χ^2^)	OR (95%CI, Wald χ^2^)
Univariate model				
Injection site reaction	0.52 (0.35–0.76, 11.23) **	2.95 (1.16–7.46, 5.21) *	0.43 (0.16–1.11, 0.49)	0.99 (0.97–1.01, 0.86)
Fatigue	0.78 (0.60–1.01, 3.63)	2.28 (1.16–4.51, 5.67) *	0.63 (0.29–1.38, 1.32)	1.01 (0.99–1.02, 0.31)
Chills	0.69 (0.53–0.90, 7.34) **	1.67 (0.84–3.33, 2.11)	0.23 (0.08–0.70, 6.81) **	1.00 (0.98–1.02, 0.02)
Fever	0.64 (0.48–0.85, 9.56) **	1.51 (0.74–3.08, 1.31)	0.19 (0.06–0.67, 6.79) **	1.00 (0.99–1.02, 0.29)
Arthralgia	0.76 (0.60–0.96, 5.28) *	1.91 (1.01–3.60, 4.00) *	0.46 (0.21–1.02, 3.68)	1.00 (0.99–1.02, 0.38)
Myalgia	0.87 (0.69–1.09, 1.40)	2.20 (1.18–4.10, 6.11) *	0.48 (0.23–1.03, 3.52)	1.01 (1.00–1.03, 2.95)
Headache	0.71 (0.56–0.89, 8.31) **	4.82 (2.50–9.28, 22.05) **	0.99 (0.48–2.02, <0.01)	1.01 (0.99–1.02, 0.69)
Diarrhea	0.52 (0.25–1.04, 3.38)	0.91 (0.20–4.18, 0.02)	0.59 (0.07–5.06, 0.23)	0.99 (0.95–1.03, 0.26)
Nausea	0.88 (0.65–1.18, 0.77)	1.64 (0.72–3.71, 1.39)	0.46 (0.15–1.40, 1.88)	1.00 (0.98–1.02, 0.05)
Any adverse event	0.26 (0.12–0.55, 12.12) **	5.58 (1.12–27.69, 4.42) *	0.12 (0.03–0.50, 8.38) **	0.99 (0.96–1.03, 0.16)
Multivariate model				
Injection site reaction	0.51 (0.33–0.79, 9.22) **	2.23 (0.83–5.98, 2.55)	0.79 (0.26–2.38, 0.17)	0.98 (0.95–1.00, 3.58)
Fatigue	0.84 (0.63–1.12, 1.44)	2.08 (1.04–4.18, 4.28) *	0.79 (0.34–1.81, 0.32)	1.00 (0.98–1.02, 0.05)
Chills	0.74 (0.55–0.98, 4.27) *	1.47 (0.71–3.02, 1.09)	0.29 (0.09–0.88, 4.76) *	1.00 (0.98–1.01, 0.23)
Fever	0.68 (0.50–0.93, 5.90) *	1.32 (0.62–2.78, 0.52)	0.25 (0.07–0.88, 4.68) *	1.00 (0.98–1.02, 0.04)
Arthralgia	0.82 (0.63–1.06, 2.38)	1.74 (0.91–3.33, 2.76)	0.55 (0.24–1.26, 1.99)	1.00 (0.99–1.02, 0.04)
Myalgia	1.01 (0.79–1.31, 0.01)	2.21 (1.16–4.21, 5.77) *	0.50 (0.22–1.11, 2.89)	1.02 (1.00–1.03, 2.96)
Headache	0.73 (0.55–0.96, 5.11) *	4.54 (2.32–8.89, 19.44) **	1.50 (0.65–3.48, 0.88)	1.00 (0.99–1.02, 0.12)
Diarrhea	0.44 (0.20–0.98, 4.03) *	0.64 (0.13–3.13, 0.31)	0.97 (0.10–8.95, <0.01)	0.97 (0.93–1.02, 1.27)
Nausea	0.93 (0.67–1.28, 0.22)	1.53 (0.67–3.52, 1.02)	0.50 (0.16–1.57, 1.42)	1.00 (0.98–1.02, 0.11)
Any adverse event	0.30 (0.13–0.73, 7.10) **	2.67 (0.45–15.83, 1.17)	0.31 (0.06–1.60, 1.95)	0.97 (0.93–1.01, 1.97)

Logistic regression was performed in both univariate and multivariate models in order to examine the association between each adverse reaction and age, sex, comorbidities, and vaccination interval. Vomiting was omitted from the analysis because only two of the vaccine recipients developed vomiting. OR: odds ratio, CI: confidential interval, *, *p* < 0.05; **, *p* < 0.01. OR and 95% CI of age represent ratios per 10 years of age. OR and 95% CI for sex indicate female/male ratios. OR and 95% CI of comorbidities show the ratio of those with/without comorbidities.

**Table 6 vaccines-10-01357-t006:** Correlation between age, vaccination interval, and severity of each adverse reaction.

	1 Age	2	3	4	5	6	7	8	9	10	11	12	13
1 Age	-												
2 Vaccination interval	−0.238 **	-											
3 Injection site reaction	−0.225 **	0.030	-										
4 Fatigue	−0.179 *	0.085	0.516 **	-									
5 Chills	−0.184 *	0.052	0.400 **	0.518 **	-								
6 Fever	−0.220 **	0.080	0.410 **	0.502 **	0.649 **	-							
7 Arthralgia	−0.151 *	0.117	0.257 **	0.542 **	0.416 **	0.413 **	-						
8 Myalgia	−0.089	0.154 *	0.290 **	0.489 **	0.356 **	0.253 **	0.645 **	-					
9 Headache	−0.187 *	0.084	0.400 **	0.568 **	0.370 **	0.373 **	0.493 **	0.389 **	-				
10 Diarrhea	−0.142	−0.031	0.264 **	0.246 **	0.188 *	0.165 *	0.066	0.168 *	0.180 *	-			
11 Nausea	−0.078	−0.002	0.240 **	0.433 **	0.306 **	0.381 **	0.403 **	0.402 **	0.413 **	0.288 **	-		
12 Vomiting	0.015	−0.005	0.158 *	0.142	0.186 *	0.199 **	0.059	0.053	0.045	−0.022	0.269 **	-	
13 Worst adverse reaction	−0.210 **	0.059	0.756 **	0.751 **	0.490 **	0.520 **	0.470 **	0.449 **	0.607 **	0.285 **	0.392 **	0.179 *	-

Spearman’s rank correlation was performed. The values indicate the correlation coefficient between age, vaccination interval, and severity of each adverse reaction. *, *p* < 0.05; **, *p* < 0.01. The most severe grade of all classified adverse reactions was defined as the worst adverse reaction grade.

## Data Availability

Data is contained within the article.

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
