# Peer review of "Young Age, Female Sex, and No Comorbidities Are Risk Factors for Adverse Reactions after the Third Dose of BNT162b2 COVID-19 Vaccine against SARS-CoV-2: A Prospective Cohort Study in Japan"

_vaccines, 2022, doi:10.3390/vaccines10081357_

Round 1
Reviewer 1 Report
1. There was a lack of information about possible severe AE (myocarditis in young adult men, other cardiovascular diseases, thrombosis). I understand that they did not exist in the study group, but it is worth supplementing it in this publication.
2. Indeed the autors should explain (I haven't found it) what it means „the worst adverse reaction”
If the study is continued, it may be possible (there is such a comment in the article) to complete the survey information from all those vaccinated with the 3rd dose.
Since vaccination with the fourth dose has already started (also in Japan), continue the study. Whether subsequent doses are associated with a higher risk of AE.
I suggest adding to the questionnaire a question about SARS infection, did they get sick despite vaccinations, if so, how many doses, at what time from vaccination, what was the course of the infection.
All this is already a proposal beyond the scope of the current publication, it concerns only a possible continuation of the study.
Author Response
Response to Reviewer 1
We would like to thank you for reading our manuscript in detail and having the opportunity to revise. We wish to express our appreciation to your insightful comments, which have helped us importantly improve the manuscript.
Comment:
There was a lack of information about possible severe AE (myocarditis in young adult men, other cardiovascular diseases, thrombosis). I understand that they did not exist in the study group, but it is worth supplementing it in this publication.
Response:
As you pointed out, we have no vaccinees who developed severe AE such as myocarditis, cardiovascular diseases and thrombosis. We inserted the following sentence in the section of Profiles of research subjects and adverse reactions of second and third doses.
Line 145-146:
“No severe AE such as myocarditis, cardiovascular diseases and thrombosis developed.”
Comment:
Indeed the authors should explain (I haven't found it) what it means „the worst adverse reaction”
Response:
We added the following explanation of “worst adverse reaction” to Materials and Methods, and table legends.
Line 114-115, 154-155, 167-168, 174-175, and 212-213:
“The most severe grade of all classified adverse reactions was defined as grade of the worst adverse reaction.”
Comment:
If the study is continued, it may be possible (there is such a comment in the article) to complete the survey information from all those vaccinated with the 3rd dose.
Since vaccination with the fourth dose has already started (also in Japan), continue the study. Whether subsequent doses are associated with a higher risk of AE.
I suggest adding to the questionnaire a question about SARS infection, did they get sick despite vaccinations, if so, how many doses, at what time from vaccination, what was the course of the infection.
All this is already a proposal beyond the scope of the current publication, it concerns only a possible continuation of the study.
Response:
Thank you for your valuable comment. We would really like to get your suggestions as we have already started the fourth round of vaccinations and are considering continuing the research.
Reviewer 2 Report
Dear authors
It was with great pleasure that I reviewed your manuscript.
However, I have some suggestions for improvement to propose:
-The Introduction should be a little more in-depth.
-The description of the sample constitution should be in the method and not in the results.
-The chi-square tests are not well described as they should have included the chi-square value and the degrees of freedom.
-The table presenting the logistic regression results is also incomplete. They should present at least the Wald chi-square value.
-The table with the Pearson correlation values is not presented according to APA 7th Edition standards.
-The tables do not have a well-done legend, also according to APA norms 7th Edition.
-All the results are described in a very superficial way. They need to be described in more detail in the body of the text.
My Best Regards
Author Response
Response to Reviewer 2
We would like to appreciate you for reviewing our manuscript and giving us the opportunity to revise it. We think your insightful comments have helped us improving our manuscript. Thank you very much for all your comments and we would appreciate if you have any further comments.
Comment:
-The Introduction should be a little more in-depth.
Response:
Thank you for your suggestion. We revised the following descriptions.
Line 46-53:
First manuscript: The more the virus replicates, the greater the risk that a variant will emerge, and that more dangerous or more easily transmitted variants may emerge in the future.
Revised manuscript: In general, the more the virus replicates, the greater the risk that a variant will emerge, and that more dangerous or more easily transmitted variants may emerge in the future. The mechanism of viral mutation is determined by multiple virus- and host-dependent processes. RNA viruses mutate faster than DNA viruses, single-stranded viruses mutate faster than double-stranded viruses, and genome size is negatively correlated with mutation rate [4]. Since SARS-CoV-2 is positive-strand RNA viruses of relatively large size, we should continue to watch for rapid mutation [5].
Also, we added the following description in Line 73-79
“These may be explained by previous reports of sex and age differences in the immune system. Sex hormones affect immune responses via secondary metabolites that bind to receptors such as estrogen receptors (ERs) and peroxisome proliferator-activated re-ceptors (PPARs) [17]. Aging brings age-related changes in the cells of the immune sys-tem, the soluble molecules that guide the maintenance and function of the immune system, and the lymphoid organs that coordinate both the maintenance of lympho-cytes and the initiation of immune responses [18].”
Comment:
-The description of the sample constitution should be in the method and not in the results.
Response:
As you pointed out, we have moved the following descriptions of 3.1 Profiles of research subjects and adverse reactions of second and third doses from Results to Materials and Methods.
Line 98-102
“A total of 617 health care workers were vaccinated for the third dose, of whom 224 consented to the study and completed the questionnaire (response rate: 36.3%). There were 194 valid responses (valid response rate: 86.6%), of which 175 who were vaccinated with all three doses of BNT162b2 were included in this study for the third vaccination.”
Comment:
-The chi-square tests are not well described as they should have included the chi-square value and the degrees of freedom.
Response:
We added chi-square value and the degrees of freedom to Tables 3 and 4. Also, we have corrected some p-value because we noticed an error in adopting the value of Fisher's exact test when we should have adopted the value of chi-square for the p-value. Thank you for giving us the opportunity to realize this mistake.
Comment:
-The table presenting the logistic regression results is also incomplete. They should present at least the Wald chi-square value.
Response:
We added the Wald chi-square value to Table 5.
Comment:
-The table with the Pearson correlation values is not presented according to APA 7th Edition standards.
Response:
We revised Table 6 according to APA 7th Edition standards.
Comment:
-The tables do not have a well-done legend, also according to APA norms 7th Edition.
Response:
We have revised all tables on the basis of your suggestion.
Comment:
-All the results are described in a very superficial way. They need to be described in more detail in the body of the text.
Response:
We have revised the results to be more specific, as you indicated.
We added the following description at Line 145-146:
“No severe AE such as myocarditis, cardiovascular diseases and thrombosis developed.”
Line 179-189:
First manuscript: The results of the logistic regression analysis are shown in Table 5. In the univariate model, significant differences were detected in 6 adverse reactions by age and sex and 3 by comorbidities. In the multivariate model, significant differences were observed in 6 adverse reactions by age, 3 by sex, and 2 by comorbidities. Consistently, the risk of developing adverse reactions was greater among the young, females, and those vaccinated without comorbidities. As for the vaccination intervals, no significant differences were detected in univariate and multivariate models.
Revised manuscript: The results of the logistic regression analysis are shown in Table 5. In the univariate model, significant differences were detected in 6 adverse reactions (injection site reaction, chills, fever, arthralgia, headache, and any AE) by age, 6 adverse reactions (injection site reaction, fatigue, arthralgia, myalgia, headache, and any AE) by sex, and in 3 adverse reactions (chills, fever, and any AE) by comorbidities. In the multi-variate model, significant differences were observed in 6 adverse reactions (injection site reaction, chills, fever, headache, diarrhea, and any AE) by age, 3 (fatigue, myalgia, and headache) and by sex, and 2 (chills and fever) by comorbidities. Consistently, the risk of developing adverse reactions was greater among the young, females, and those vaccinated without comorbidities. As for the vaccination intervals, no significant differences were detected in both univariate and multivariate models.
Line 202-208
First manuscript: Correlation coefficients between age and vaccination interval and severity of ad-verse reactions are shown in Table 6. Younger age was significantly correlated with the severity of many adverse reactions. Only myalgia showed a weak correlation be-tween vaccination interval and severity of adverse reactions, but no other significant correlation was observed. A significant negative correlation was detected between age and vaccination interval.
Revised manuscript: Correlation coefficients between age and vaccination interval and severity of ad-verse reactions are shown in Table 6. Younger age was significantly correlated with the severity of many adverse reactions (injection site reaction, fatigue, chills, fever, arthralgia, headache and the worst adverse reaction). Only myalgia showed a weak correlation between vaccination interval and severity of adverse reactions, but no other significant correlation was observed. A significant negative correlation was de-tected between age and vaccination interval.
Round 2
Reviewer 2 Report
Dear authors
Thank you for following all the suggestions I gave you to improve your manuscript.
My Best Regards